# Feed Restriction Reveals Distinct Serum Metabolome Profiles in Chickens Divergent in Feed Efficiency Traits

**DOI:** 10.3390/metabo9020038

**Published:** 2019-02-25

**Authors:** Barbara U. Metzler-Zebeli, Sina-Catherine Siegerstetter, Elizabeth Magowan, Peadar G. Lawlor, Niamh E. O’Connell, Qendrim Zebeli

**Affiliations:** 1Institute of Animal Nutrition and Functional Plant Compounds, Department for Farm Animals and Veterinary Public Health, University of Veterinary Medicine Vienna, 1210 Vienna, Austria; sina-catherine.siegerstetter@vetmeduni.ac.at (S.-C.S.); qendrim.zebeli@vetmeduni.ac.at (Q.Z.); 2Agri-Food and Biosciences Institute, Agriculture Branch, Hillsborough, Northern Ireland BT26 6DR, UK; Elizabeth.Magowan@afbini.gov.uk; 3Teagasc, Pig Development Department, Animal & Grassland Research & Innovation Centre, Moorepark, Co. Cork, P61 C996 Fermoy, Ireland; Peadar.Lawlor@teagasc.ie; 4Institute for Global Food Security, Queen’s University Belfast, Belfast BT7 1NN, UK; niamh.oconnell@qub.ac.uk

**Keywords:** broiler chicken, serum metabolome, amino acids, lipids, feed intake level, residual feed intake

## Abstract

Restrictive feeding influences systemic metabolism of nutrients; however, this impact has not been evaluated in chickens of diverging feed efficiency. This study investigated the effect of ad libitum versus restrictive feeding (85% of ad libitum) on the serum metabolome and white blood cell composition in chickens of diverging residual feed intake (RFI; metric for feed efficiency). Blood samples were collected between days 33 and 37 post-hatch. While serum glucose was similar, serum uric acid and cholesterol were indicative of the nutritional status and chicken’s RFI, respectively. Feed restriction and RFI rank caused distinct serum metabolome profiles, whereby restrictive feeding also increased the blood lymphocyte proportion. Most importantly, 10 amino acids were associated with RFI rank in birds, whereas restrictive feeding affected almost all detected lysophosphatidylcholines, with 3 being higher and 6 being lower in restrictively compared to ad libitum fed chickens. As indicated by relevance networking, isoleucine, lysine, valine, histidine, and ornithine were the most discriminant for high RFI, whereas 3 biogenic amines (carnosine, putrescine, and spermidine) and 3 diacyl-glycerophospholipids (38:4, 38:5, and 40:5) positively correlated with feed intake and body weight gain, respectively. Only for taurine, feed intake mostly explained the RFI-associated variation, whereas for most metabolites, other host physiological factors played a greater role for the RFI-associated differences, and was potentially related to insulin-signaling, phospholipase A2, and arachidonic acid metabolism. Alterations in the hepatic synthesis of long-chain fatty acids and the need for precursors for gluconeogenesis due to varying energy demand may explain the marked differences in serum metabolite profiles in ad libitum and restrictively fed birds.

## 1. Introduction

An efficient conversion of nutrients into sellable meat is an important aspect for the sustainable intensification of chicken production [1]. Due to the increasing production costs and the demand to reduce the environmental impact of poultry production, the search for selection strategies to accurately predict feed efficiency (FE) has intensified [1]. As shown in the past, simple ratio metrics (e.g., gain:feed ratio) do not take into account individual variation in metabolizable energy requirements. Therefore, as a heritable trait and being independent of production traits, the residual feed intake (RFI) has become the metric of choice for studying physiological mechanisms underlying the variation of FE in poultry [1,2,3]. The RFI divides the feed intake into maintenance and production components. Linear regression is used to compute the expected feed intake based on mid-test metabolic body weight and growth in broiler chickens, with the difference between the actual and expected feed intake being defined as RFI [1]. This means that the RFI corresponds to the feed consumed above (positive RFI value indicating low FE) or below (negative RFI indicating high FE) the expected feed intake needed to meet the energy and nutrient requirements for a certain production level [1]. Accordingly, the major phenotypic differences between chickens of divergent RFI are the feed intake level (FL), which is accompanied by RFI-associated variation in gut microbiota and physiology, as well as in energy metabolism [2,3]. Understanding the mechanisms that underlie individual animal variation is therefore essential for the development of strategies to improve the FE in poultry. Serum intermediary metabolites, i.e., uric acid and cholesterol, may be used as FE predictors [4] in that they support FE-related differences in nutrient assimilation, and hepatic, muscle, and adipocyte metabolism [5]. Similar differences in serum profiles were reported for the restrictive feeding of chickens, which has the goal of preventing metabolic diseases, such as fatty liver, and reduce abdominal fat accretion [6]. This has become necessary because commercial broiler lines accumulate excessive adipose tissue as a result of genetic selection for growth and being naturally insulin resistant, resembling metabolic conditions of obesity and type-2-diabetes in humans [7]. Lowering the FL shifts the metabolism of nutrients towards gluconeogenesis, thereby reducing nutrient availability for immune function [7]. 

For the successful use as FE predictors, it is essential to understand to what extent blood metabolites are related to the chicken’s feeding behavior or to physiological differences. Precise quantitative restriction can be used to ensure that all birds eat the same amount of feed, which theoretically creates similar intestinal and systemic nutrient profiles irrespective of the bird’s FE. We hypothesized that FL in birds can explain some FE-associated variation in serum metabolites. Therefore, the present objective was to investigate the effect of ad libitum versus restrictive feeding (85% of ad libitum) on the serum metabolome and white blood cells in broiler chickens of diverging RFI.

## 2. Materials and Methods 

### 2.1. Ethical Statement

All experimental procedures including animal handling and treatment were approved by the institutional ethics committee of the University of Veterinary Medicine Vienna and the Austrian national authority according to paragraph 26 of Law for Animal Experiments, Tierversuchsgesetz 2012—TVG 2012 (GZ 68.205/0148-II/3b/2015).

### 2.2. Experimental Design and Determination of Feed Efficiency

The detailed description of the experimental design, and housing and environmental conditions can be found in Siegerstetter et al. [8]. Cobb 500 female (*n* = 57) and male broiler chicks (*n* = 55), 1 day of age, were used in two consecutive replicate batches (56 chickens per batch), with one more female and one less male in batch 1 in comparison to batch 2. All chickens had free access to demineralized water from manual drinkers and were fed corn-soybean meal-based starter, grower, and finisher diets (Appendix A), which were free of antibiotics and coccidiostats. 

To ensure sufficient feed intake in the first days of life, chicks (*n* = 5–6) of the same sex were group-housed in stainless steel metabolic cages from 1 to 8 days post-hatch (dph), where all chicks had ad libitum access to feed. On 9 dph, the birds were randomly assigned to one of two treatments and their individual feed intake was recorded until the end of the experimental period (33–37 dph). Half of the chickens had ad libitum access to feed (*n* = 29 females and *n* = 28 males across both replicate batches), whereas the other half were restrictively fed (for both replicate batches together: *n* = 28 females and *n* = 27 males). The daily feed allocation of restrictively fed chickens was aimed to correspond to 90–95% of the average daily ad libitum feed intake observed in a previous chicken trial in which we assessed the RFI of the same chicken line [9]. A further aim was that all chickens in the restrictive feeding group should eat the same amount of feed and that the feeders of the restrictively fed chickens were empty the next morning. Considering that females eat less than males [9], the feed amount for the restrictively fed chickens was adjusted daily towards the female and male chicken with the lowest feed intake on the day before. For both treatment groups, fresh feed was provided at 9:00 h, and feeders were refilled at 15:00 h. 

Chickens were weighed on 1, 7, 9, 14, 21, 28, and 30 dph and the feed intake of each bird was recorded on 9, 14, 21, 28, and 30 dph. Feed refusals were collected daily before morning feeding, and feed spills were collected weekly. The RFI as metric for FE was determined for the experimental period between 9 and 30 dph. The total feed intake (TFI), metabolic mid-test BW (MMW), and total body weight gain (TBWG) were used to estimate the chickens’ RFI as the residuals over the test period (9 to 30 dph) using a nonlinear mixed model (SAS Stat Inc., version 9.4; Cary, NC, USA) [9]. In each replicate batch, separately for females and males and balanced for batch, the chickens with the lowest RFI (good FE) and highest RFI (poor FE) values in each FL group were selected to be sampled. Across both batches, 14 low RFI (*n* = 7 per sex) and 15 high RFI (*n* = 8 females and *n* = 7 males) ad libitum fed chickens, and 14 low RFI (*n* = 7 per sex) and 14 high RFI (*n* = 7 per sex) restrictively fed chickens were chosen for the measurement of serum parameters. 

Chickens were euthanized with an overdose of Thiopental (50–100 mg/kg, medicamentum pharma GmbH, Allerheiligen im Mürztal, Austria) via i.v. injection into the caudal tibial vein in the morning from 33 to 37 dph (*n* = 6/day; except on 37 dph with *n* = 4). Immediately after the injection, blood samples from the vena jugularis were collected into serum tubes (Sarstedt, Nürnberg, Germany). Blood samples were kept on ice until centrifugation at 1811× *g* for 10 min (Eppendorf Centrifuge 5810 R, Eppendorf, Hamburg, Germany). Serum aliquots were stored at −80 °C until analysis. In addition, 1 mL blood was collected into tubes containing EDTA as anticoagulant (Sarstedt, Nürnberg, Germany) from which blood smears were prepared on glass slides (*n* = 4/chicken) to count white blood cells.

### 2.3. Blood Leukocyte Counts and Clinical Biochemistry 

Blood smears were stained using the May–Grünwald–Giemsa stain (Hemacolor Rapid staining of blood smear kit; Merck KGaA, Darmstadt, Germany) [4], following which 100 leukocytes, including granular (heterophils, eosinophils, and basophils) and nongranular (lymphocytes and monocytes), were counted per slide using light microscopy (Leitz Orthoplan, Leitz, Wetzlar, Germany) at 100-times magnification. The heterophil:lymphocyte ratio was calculated [10]. Serum glucose, uric acid, triglycerides, cholesterol, and non-esterified fatty acids (NEFA) were determined using standard enzymatic colorimetric analysis using an autoanalyzer for clinical chemistry (Cobas 6000/c501; Roche Diagnostics GmbH, Vienna, Austria). 

### 2.4. Serum Metabolomics

Serum metabolites were analyzed and quantified using a targeted metabolomics approach with the AbsoluteIDQ p180 kit (BIOCRATES Life Sciences AG, Innsbruck, Austria) using the TargetIDQ Service of BIOCRATES Life Science AG as described in Metzler-Zebeli et al. [11]. This kit is based on electrospray ionization liquid chromatography–mass spectrometry, and allows for the simultaneous identification and quantification of 188 endogenous metabolites from 10 μL of serum, including amino acids (*n* = 21), biogenic amines (*n* = 21), acylcarnitines (*n* = 40), diacyl (*n* = 38) and acyl-alkyl (*n* = 38) phosphatidylcholines, lysophosphatidylcholines (*n* = 14), sphingomyelins (*n* = 15), and sum of hexoses (*n* = 1). The method of the AbsoluteIDQ™ p180 kit has been demonstrated to be in conformance with the FDA-Guideline “Guidance for Industry - Bioanalytical Method Validation (May 2001),” which implies proof of reproducibility within a given error range. Measurements were performed as described in the manufacturer’s manual UM-P180. The p180 kit is fully automated and run with internal standards that are used as references to calculate metabolite concentrations. Identification of targeted metabolites was performed using mass spectrometry (4000 QTRAP system; Applied Biosystems/MDS Sciex, Foster City, CA, USA). Chromatograms were analyzed by the BIOCRATES Life Science AG.

### 2.5. Statistical Analysis

After analyzing for normality using the Shapiro–Wilk test (SAS Stat Inc., version 9.4; Cary, NC, USA), FE parameters, serum metabolites, and the proportional composition of white blood cells were subjected to ANOVA using the MIXED procedure in SAS. The fixed effects of batch, sex, FL, RFI, and the two-way-interaction FL × RFI were considered in the main model. Batch was considered as random effect in the final model. Chicken nested within batch was the experimental unit. Degrees of freedom were approximated using the Kenward–Roger method. Differences among least squares means were computed using the pdiff statement. Differences were considered significant if *p* ≤ 0.05. 

To identify the most discriminant serum metabolites for FL and chicken’s RFI rank, we first classified samples using unclassified principal component analysis (PCA), which was complemented by supervised sparse partial least-squares (sPLS) regression by means of the package ‘mixOmics’ (version 6.3.2) in R studio (version 1.0.136) [12,13]. Relevance network graphs from sPLS were obtained via the function network to integrate serum metabolomics data with results for TFI, TBWG, and RFI. Only the strongest associations are presented.

## 3. Results

### 3.1. Restrictive Feeding Improves Residual Feed Intake

Chickens with extremely low and high RFI values were selected in both FL groups, showing contrasting RFI values as reported previously [8]. Due to the importance of the feed intake and performance data for the understanding of the current effects on the serum metabolome, these results can be found in Appendix A. In brief, restrictive feeding decreased the RFI value of high-RFI females by 153 g compared to ad libitum-fed high-RFI females (FL × RFI; *p* < 0.05). Although post-hoc comparisons showed a similar trend (*p* < 0.10) for males, the FL × RFI interaction did not reach significance (FL × RFI; *p* = 0.171). Restrictively fed chickens ate on average 338 g less than ad libitum fed birds (*p* < 0.05). Low-RFI chickens commonly ate less than high-RFI chickens [9], which resulted in a less strict feed restriction in the low-RFI chickens (92% of ad libitum group) compared to high-RFI birds (80% of ad libitum group). 

### 3.2. Serum Metabolome

The targeted metabolomics approach provided results for 139 metabolites (Appendix A). These were comprised the sum of hexoses, 35 amino acids and biogenic amines, 7 acylcarnitines, 14 sphingolipids and 82 phospholipids, including 12 lysophosphatidylphosphates and 70 glycerophospholipids. Although the sum of hexoses was similar among chicken groups in the postprandial phase (Appendix A), feed restriction increased (*p* < 0.05) the serum concentration of 4 amino acids (asparagine, citrulline, proline, and serine) and 4-hydroxyproline, and decreased the concentration of the biogenic amines carnosine and taurine compared to ad libitum feeding (*p* < 0.05; Table 1). In total, 10 amino acids were associated with RFI rank in birds, with histidine, isoleucine, leucine, lysine, ornithine, proline, serine, threonine, and valine being lower and tyrosine being higher in low RFI compared to high-RFI chickens (*p* < 0.05). Likewise, carnosine and sarcosine were higher and lower, respectively, in low RFI versus high-RFI chickens (*p* < 0.05). The FL × RFI interactions (*p* < 0.05) for glycine and taurine showed that RFI-associated differences were only found in the restrictively fed (glycine) or ad libitum fed group (taurine). Moreover, the concentration of symmetric dimethylarginine was decreased with restrictive feeding compared to ad libitum feeding, however only in low RFI but not in high-RFI chickens as indicated by the FL × RFI interaction (*p* = 0.040).

Regarding mitochondrial fatty acid oxidation, acylcarnitines were unaffected by FL; however, hexadecanoylcarnitine was associated with RFI rank, being 12.2% higher (*p* = 0.028) in high compared to low-RFI chickens (Table 2). By contrast, restrictive feeding altered (*p* < 0.05) the serum concentrations of almost all detected lysophosphatidylcholines, with 3 being higher and 6 being lower in restrictively compared to ad libitum fed chickens (Table 2). Concurrently, 4 lysophosphatidylcholines were higher (*p* < 0.05) in high- versus low-RFI chickens. The concentration of sphingomylein C24:0 was higher in high- versus low-RFI chickens, whereas restrictive feeding did not change serum sphingomyelin concentrations (Table 2). 

From the glycerophospholipids, restrictive versus ad libitum feeding increased and decreased serum concentrations of 6 diacyl-glycerophosphatidylcholines, respectively, as well as decreased 6 acyl-alkyl-glycerphosphatidylcholines (*p* < 0.05; Table 3). Moreover, 6 and 4 diacyl- glycerophosphatidylcholines were higher and lower, respectively, in high compared to low-RFI birds (*p* < 0.05). Additionally, 3 acyl-alkyl-glycerophosphatidylcholines were higher (*p* < 0.05) in high- versus low-RFI chickens.

### 3.3. Serum Biochemistry and White Blood Cell Counts 

Serum biochemistry showed lower serum uric acid (*p* = 0.024) in restrictively compared to ad libitum fed chickens (Table 4; Appendix A). By contrast, feed restriction increased (*p* = 0.038) the percentage of lymphocytes by 4.6% compared to ad libitum feeding (Table 4; Appendix A). Serum cholesterol was 10.2% higher (*p* = 0.040) in high- versus low-RFI chickens.

### 3.4. Supervised Data Integration: Serum Predictor Identification

In order to identify the most influential serum metabolites for feed intake, growth performance, and RFI, we first used unsupervised PCA to classify samples that demonstrated similarity in the serum metabolome profiles among all four chicken groups (data not shown). We further tuned the metabolome data by performing sPLS regression, first retaining one-quarter of the metabolome variables and later tuning the number of retained variables to one-tenth of all metabolites. Scatter plots for the identified serum metabolites showed overlapping 95%-confidence intervals (Figure 1). 

Through the loadings for component 1, 7 glycerophospholipids, 1 lysophospholipid, and 4 biogenic amines and citrulline were identified as the most discriminant metabolites (Figure 2A), with the diacyl-glycerphosphatidylcholine C38:4 being the most explicative parameter. In addition, relevance networks, inferred from pairwise association scores between X (metabolites) and Y (performance) variables, were built to reveal associations between metabolites, RFI, TFI, and TBWG (Figure 2B). Five amino acids (isoleucine, lysine, valine, histidine, and ornithine) were positively associated (r > 0.4) with the individual RFI values, whereas 3 biogenic amines (carnosine, putrescine, and spermidine) were the best discriminants for TFI and 3 diacyl-glycerophospholipids (C38:4, C38:5, and C40:5) for TBWG.

## 4. Discussion

The present results demonstrated distinct serum metabolite profiles in low and high-RFI chickens. Therefore, metabolic adaptations caused by the manipulation of energy and nutrient supply due to restrictive feeding was expected in these animals, which would allow for differentiation between FL effects and effects due to variation in host physiology on serum metabolites. Precise quantitative feed restriction ensured the intake of equal amounts of nutrients by all females and males, which theoretically should result in similar systemic nutrient supply regardless of a chicken’s FE. The present data confirm our hypothesis that for certain serum amino acids, taurine, and lyso- and diacyl-glycerophosphatidylcholines, the FL helped to explain in part the RFI-variation observed. Moreover, serum profiles allowed for the identification of metabolite patterns that were either characteristic for restrictive feeding or bird’s RFI rank. Elevated levels of serum histidine, isoleucine, lysine, valine, and ornithine were thereby indicative of high RFI, whereas biogenic amines (i.e., spermidine, carnosine, and putrescine), taurine, diacyl-glycero- and lysophosphatidylcholines were characteristic of the chicken’s nutritional status as identified in the loadings and relevance networks. By contrast, serum sphingomyelin and triglyceride levels in the postprandial state did not predict RFI rank or FL. Previously, we identified serum uric acid and cholesterol as potential predictors for RFI in chickens [4]. The present results support the applicability of cholesterol as a predictor of RFI, whereas serum uric acid was mainly indicative of the nutritional status of the chicken. Probably linked to the lower energy and amino acid availability, restrictive feeding but not chicken’s RFI altered the proportional composition of white blood cells. Since we only measured the proportional composition, no conclusions can be drawn as to whether restrictive feeding or RFI rank affected the chicken’s immune status. Overall, the present results support RFI-mediated effects via intestinal absorption (e.g., amino acid transporters, bile secretion, or micelle formation), hepatic lipoprotein formation, and muscle and adipocyte metabolism [5].

Although restrictive feeding successfully improved the FE in high-RFI birds, low and high-RFI ranked chickens still had distinctly different RFI values, indicating that host physiological differences persisted [8]. This may be the underlying reason for the lack of sample separation in the scatter plots along the most explicative component 1 and 2. Moreover, despite the lower daily allocation of feed for the restrictively fed chickens, feed was available to chickens for most of the day, with birds only fasted from either the late evening or early morning before the feeders were refilled. Consequently, we predict that the restrictively fed chickens may have maintained a narrower insulin:glucagon ratio [14] throughout the day compared to ad libitum fed birds. As low-RFI chickens eat less feed than their high-RFI counterparts [9], the fasting period was shorter for the restrictively fed low-RFI birds than for the restrictively fed high-RFI animals, probably leading to fewer metabolic adaptations and closer metabolite profiles between low-RFI birds of the ad libitum and restrictively fed groups.

In the fasting state, hormonal and metabolic signaling diverts nutrients for glyceroneogenesis and ATP generation by up-regulating proteolysis and amino acid catabolism, as well as fatty acid mobilization and oxidation [7]. Since the chickens in the present study were sampled in the postprandial phase, glucose provided by the feed was oxidized for energy, thereby prompting insulin-dependent signalling and anabolism. Despite equal postprandial glucose levels between FL groups, serum data indicated continuing alterations in the protein and lipid metabolism in response to lower feed availability in restrictively fed chickens. Previously, restrictively feeding chickens a corn-soybean meal-based diet, as was the case in the present study, up-regulated the peptide transporter in the small intestine [15]. This may support our findings for elevated serum asparagine, proline, and serine. However, fasting has also been shown to slow down muscle protein accretion in chickens during the re-feeding period [16], being another valid explanation for the altered serum amino acid levels in the present study. Differences in the catabolism of amino acids may have been indicated by the elevated level of citrulline in restrictively versus ad libitum fed chickens. By being produced by enterocytes in the small intestine during the oxidation of proline [17], plasma citrulline is indicative of acute and chronic gut integrity loss in humans [18]. Restrictive feeding, however, did not impair jejunal barrier function in the present study [19]. Since both proline and citrulline were elevated, adjustments in mucosal amino acid oxidation in favor of hepatic and muscle protein synthesis is likely. As 4-hydroxyproline, with which proline forms the two major amino acids for collagen protein synthesis [17], was raised in response to restrictive feeding, this could indicate a shift in protein anabolism toward joint tissue formation in restrictively fed chickens. Concurrently, amino acid-saving mechanisms were indicated by the reduced serum levels of uric acid, taurine, and carnosine in restrictively fed birds, probably caused by a glucagon-related increase in proteolysis and the use of amino acids for gluconeogenesis during the fasting state. With taurochenodeoxycholate being the major avian bile salt [20], restrictive feeding may have modified the bird’s capacity to digest fat. Moreover, specific adaptive mechanisms in restrictively fed birds to spare nitrogen were indicated by similar serum uric acid concentrations across RFI ranks but greater nitrogen retention in high-RFI chickens [8]. This may have been a compensation for the greater amino acid requirement of high-RFI chickens and may have involved a greater intestinal absorption and hepatic and renal metabolism, leading to similar metabolite profiles in the restrictively and ad libitum fed chickens. Only for glycine was an FL × RFI interactive effect observed, which may have been indicative of increased gluconeogenesis in restrictively fed high- versus low-RFI chickens.

Serum amino acids that showed clear RFI-associated patterns included many proteinogenic and essential amino acids, such as histidine, isoleucine, leucine, lysine, proline, serine, threonine, valine, ornithine, and sarcosine which increased in concentration in high-RFI chickens. Only tyrosine and carnosine serum concentrations were lower in high-RFI chickens across both FL groups. However, it was only for proline, serine, and taurine that the RFI-associated variation was linked to chicken’s FL. According to the FL × RFI interaction, FL was the only main influencing factor for serum taurine concentration, whereas for the other amino acids, differences in host metabolism were stronger contributors to the observed RFI-associated variation. The higher serum amino acid concentrations in high-RFI chickens may indicate reduced peripheral amino acid utilization for skeletal muscle synthesis and differences in the disposal of amino acids via the urea cycle. These findings may be related to lower insulin sensitivity of tissues as reported previously for the breast muscle of low FE chickens [21]. Elevated plasma levels of branched-chain amino acids have been linked to the onset of type-2-diabetes and cardiovascular diseases [22], highlighting the resemblance of these patho-physiological conditions and the high RFI phenotype in the present study. The higher leucine levels in high-RFI birds may be related to changes in the postprandial stimulation of protein synthesis through cellular mammalian target of rapamycin (mTOR) signaling [17], which may be an adaptation to compensate for the lower expression of genes for cell growth and development in the breast muscle as previously observed for low FE birds [23]. Moreover, higher serum serine concentrations in high- versus low-RFI birds may indicate the upregulation of gluconeogenesis, one-carbon metabolism, and purine and uric acid synthesis [17].

Notably, serum carnosine was lower in the high-RFI phenotype and further decreased by restrictive feeding. Despite this, the underlying physiological reasons for this finding may differ and may be linked to one of the metabolic roles of carnosine, which is scavenging of reactive oxygen species (ROS) produced during mitochondrial oxidative phosphorylation and peroxidation of membrane lipids [24]. Simultaneously, in response to oxidative stress, phosphatidylcholines of low-density lipoproteins are converted into lysophosphatidylcholines via the action of phospholipase A2 [25], whereby saturated lysophosphatidylcholines have strong pro-inflammatory abilities. Hence, the lower carnosine level in restrictively fed birds may reflect a reduced requirement for ROS scavengers, which is supported by the lower serum levels of saturated lysophosphatidylcholines (i.e., lysophosphatidylcholines C17:0, C26:0, and C28:0) in restrictively compared to ad libitum fed chickens. In this context, sPLS regression emphasized that particularly the lower serum level of the polyunsaturated lysophosphatidylcholine 20:4 may be characteristic of the nutritional status of restrictively fed chickens. In line with this, high FE chickens (which correspond to the low-RFI animals in the present study) were previously characterized as having lower mitochondrial ROS and oxidative stress levels [23,26] and this is supported by the lower serum lysophosphatidylcholine and higher carnosine levels in low versus high-RFI chickens in the present study. Furthermore, the higher serum hexadecanoylcarnitine level in high- versus low-RFI birds may reflect increased mitochondrial fatty acid oxidation and subsequently increased energy production from β-oxidation, leading to an increased mitochondrial ROS production in high RFI chickens. Assuming higher insulin levels in high- versus low-RFI chickens, this may have induced greater phospholipase A2 activity in high-RFI chickens, thereby further increasing lysophosphatidylcholine levels and ROS production, while altering the glycerophospholipid composition [27].

The FL can help explain the higher serum levels of lysophosphatidylcholines C16:1, C18:1, and C20:3, as well as the higher serum levels of diacyl-glycerophosphatidylcholines C32:1, C34:1, C36:0, and C36:1, and lower levels of C40:5 in high-RFI chickens. However, as RFI-related profiles in phospholipids remained discernible in both FL groups, other host physiological factors played a role and may have been related to RFI-related differences in phospholipase A2 and arachidonic acid metabolic pathways [28]. By contrast, the RFI-variation in acyl-alkyl-phosphatidylcholines was totally independent of the chicken’s FL. Alterations in the hepatic synthesis of long-chain fatty acids due to varying energy supply and hormonal signaling may explain the changes in diacyl- and acyl-alkyl-phosphatidylcholine profiles in ad libitum and restrictively fed birds. Finally, it is interesting to note that serum diacyl-glycerophosphatidylcholines C32:1, C36:1, and C38:3, which were elevated in high-RFI chickens, belong to those phospholipids that are used as biomarkers for type-2-diabetes in humans [29].

## 5. Conclusions

These data showed that quantitative feed restriction and the RFI rank in chickens were responsible for alterations in serum amino acid, lysophosphatidylcholines, and diacyl- and acyl-alkyl-glycerophospholipid profiles. As well as cholesterol being confirmed as a predictor for RFI and uric acid reflecting the chicken’s nutritional status, relevance networking identified additional serum predictors for RFI (isoleucine, lysine, valine, histidine, and ornithine) and feed intake and growth (carnosine, putrescine, spermidine, and diacyl-glycerophospholipids C38:4, C38:5, and C40:5). Results also indicated that feed intake in chickens could explain the RFI-associated variation in serum taurine, whereas other host physiology- and metabolism-related factors (e.g., insulin-signaling, muscle metabolism, phospholipase A2 and arachidonic acid metabolism) were more influential regarding the RFI-associated variation observed for other serum metabolites. Nutrient and energy shortage in response to restrictive feeding may have caused alterations in the hepatic synthesis of long-chain fatty acids and increased the requirement for precursors for gluconeogenesis. Due to the resemblance between the current metabolic phenotypes and human nutritional diseases, i.e., obesity and type-2-diabetes, these findings may also have relevance for human health conditions.

## Figures and Tables

**Figure 1 metabolites-09-00038-f001:**
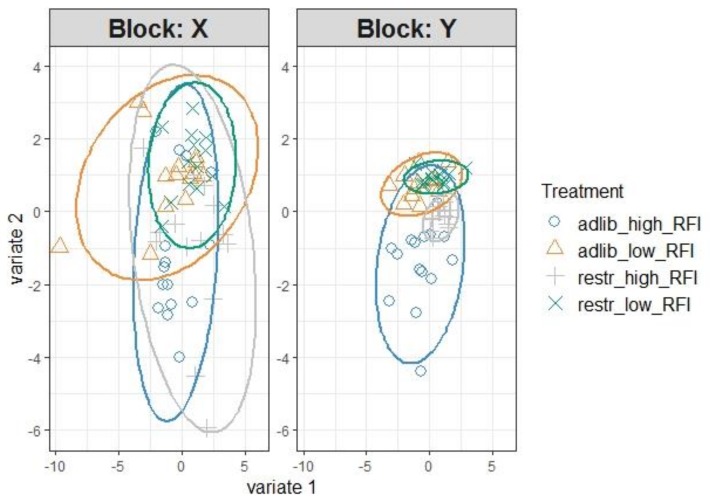
Sample plot of sparse partial least-squares regression with 95% confidence ellipse plots of serum metabolome (Block: X) and total feed intake, total body weight gain, and residual feed intake (RFI; Block: Y) of low and high residual feed intake (RFI) broiler chickens fed either ad libitum or restrictively. adlib, ad libitum feeding; restr, restrictive feeding.

**Figure 2 metabolites-09-00038-f002:**
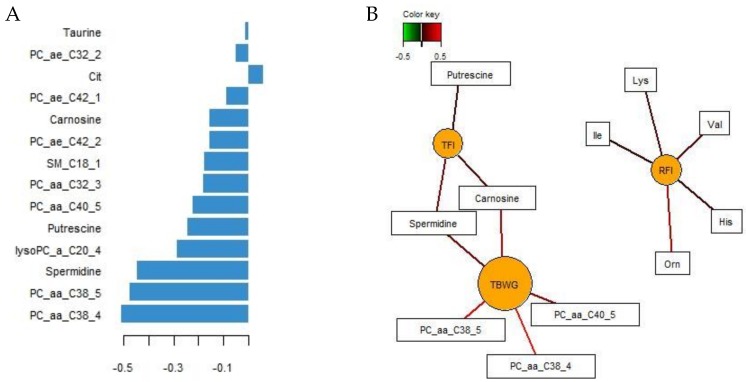
Determination of the most discriminant serum metabolites of low and high residual feed intake (RFI) broiler chickens fed either ad libitum or restrictively. (**A**) Loading plot of the most discriminant serum metabolites selected on the first component using sparse partial least-squares regression. (**B**) Relevance networks illustrating pairwise associations between serum metabolites and residual feed intake (RFI), total feed intake (TFI), and total body weight gain (TBWG;|r| > 0.4). Covariations were calculated using sparse partial least squares regression. The relevance network is displayed graphically as nodes (serum metabolites, performance traits, and RFI) and edges (biological relationship between nodes). The edge color intensity indicates the level of the association: red = positive, and green = negative. Cit, citrulline; His, histidine; Ile, isoleucine; Lys, lysine; lysoPC_a_C, with acyl residue sum C; Orn, ornithine; PC_aa_C, phosphatidylcholine with diacyl residue sum C; PC_ae_C, phosphatidylcholine with acyl-alkyl residue sum C; Val, valine.

**Table 1 metabolites-09-00038-t001:** Selected serum amino acids and biogenic amines (µmol/L) in low and high residual feed intake (RFI) broiler chickens fed either ad libitum or restrictively.

	Ad Libitum Feeding		Restrictive Feeding			*p*-Value		
Metabolite	Low RFI	High RFI	Low RFI	High RFI	SEM	FL	RFI	FL × RFI
Amino Acids								
Asparagine	108.7	86.5	133.2	153.6	15.27	0.004	0.952	0.169
Citrulline	8.1	8.7	12.0	13.4	1.03	<0.001	0.345	0.710
Glycine	555.4 ^ab^	527.6 ^ab^	481.8 ^b^	585.9 ^a^	31.87	0.810	0.233	0.042
Histidine	91.6	111.5	80.6	127.2	6.99	0.733	<0.001	0.062
Isoleucine	124.9	145.2	120.0	158.3	7.19	0.570	<0.001	0.215
Leucine	292.8	316.2	287.9	344.7	13.27	0.379	0.004	0.214
Lysine	365.8	427.6	337.6	429.7	27.57	0.638	0.007	0.586
Ornithine	41.0	56.0	38.2	65.9	5.15	0.495	<0.001	0.228
Proline	446.2	474.5	466.5	573.8	20.88	0.006	0.002	0.064
Serine	669.1	714.5	701.3	848.0	33.56	0.017	0.006	0.137
Taurine	124.3 ^b^	169.8 ^a^	121.4 ^b^	105.1 ^b^	13.28	0.014	0.275	0.024
Threonine	464.8	536.3	483.0	566.4	28.67	0.403	0.009	0.835
Tryptophan	85.6	87.2	83.8	94.8	3.29	0.378	0.059	0.159
Tyrosine	265.8	219.1	257.8	241.5	13.86	0.606	0.027	0.277
Valine	199.5	232.4	188.2	257.1	11.49	0.566	<0.001	0.123
**Biogenic Amines**								
Carnosine	21.6	16.6	15.8	12.5	1.74	0.006	0.023	0.622
Methionine sulphoxide	11.8	12.5	11.6	13.2	0.59	0.726	0.054	0.433
Sarcosine	18.3	21.6	17.7	22.2	1.20	0.987	0.002	0.633
Symmetric dimethylarginine	0.88 ^a^	0.80 ^ab^	0.78 ^b^	0.84 ^ab^	0.03	0.348	0.662	0.040
Spermidine	0.36	0.31	0.25	0.26	0.04	0.067	0.600	0.436
4-Hydroxyproline	155.0	130.0	157.7	172.0	10.50	0.038	0.615	0.067

Data are presented as least-square means and pooled SEM. RFI, residual feed intake; FL, feed intake level. *n* = 7 per FL group, RFI rank, and sex; except for *n* = 8 high-RFI ad libitum females. Means within a row with different superscripts differed significantly (*p* < 0.05). RFI was calculated for the experimental period from 9 to 30 days post-hatch.

**Table 2 metabolites-09-00038-t002:** Selected serum acyl-carnitine, lysophosphatidylcholines, and sphingomyelin (µmol/L) in low and high residual feed intake (RFI) broiler chickens fed either ad libitum or restrictively.

	Ad Libitum Feeding		Restrictive Feeding			*p*-Value		
Metabolite ^1^	Low RFI	High RFI	Low RFI	High RFI	SEM	FL	RFI	FL × RFI
Hexadecanoylcarnitine	0.022	0.024	0.019	0.022	0.001	0.094	0.028	0.803
LysoPC a C16:1	1.00	1.19	1.26	1.54	0.07	<0.001	0.002	0.506
LysoPC a C17:0	0.22	0.21	0.20	0.19	0.01	0.028	0.461	0.930
LysoPC a C18:0	23.6	27.6	25.5	28.1	1.25	0.362	0.012	0.558
LysoPC a C18:1	7.65	9.10	9.01	11.05	0.47	<0.001	<0.001	0.534
LysoPC a C18:2	13.1	14.0	13.9	15.3	0.69	0.141	0.105	0.705
LysoPC a C20:3	1.16	1.48	1.35	1.74	0.10	0.025	<0.001	0.756
LysoPC a C20:4	4.90	4.88	4.21	4.12	0.32	0.029	0.862	0.923
LysoPC a C26:0	0.13	0.10	0.08	0.07	0.02	0.013	0.182	0.326
LysoPC a C26:1	0.098	0.062	0.050	0.045	0.012	0.010	0.087	0.202
LysoPC a C28:0	0.19	0.16	0.14	0.12	0.02	0.026	0.172	0.636
LysoPC a C28:1	0.18	0.13	0.10	0.08	0.02	0.005	0.115	0.334
Sphingomyelin C24:0	12.0	13.6	13.2	15.0	0.733	0.090	0.022	0.866

Data are presented as least-square means and pooled SEM. RFI, residual feed intake; FL, feed intake level. *n* = 7 per FL group, RFI rank, and sex; except for *n* = 8 high-RFI ad libitum females. RFI was calculated for the experimental period from 9 to 30 days post-hatch. ^1^ LysoPC a C, with acyl residue sum C.

**Table 3 metabolites-09-00038-t003:** Selected serum diacyl- and acyl-alkyl-glycerophosphatidylcholines (µmol/L) in low and high residual feed intake (RFI) broiler chickens fed either ad libitum or restrictively.

	Ad Libitum Feeding		Restrictive Feeding			*p*-Value		
Metabolite ^1^	Low RFI	High RFI	Low RFI	High RFI	SEM	FL	RFI	FL × RFI
PC aa C24:0	0.14	0.10	0.07	0.06	0.017	0.003	0.147	0.371
PC aa C30:2	0.079	0.050	0.039	0.029	0.011	0.006	0.076	0.371
PC aa C32:0	26.7	26.3	27.7	28.0	1.452	<0.001	0.967	0.828
PC aa C32:1	8.6	10.2	12.5	16.9	1.160	<0.001	0.012	0.224
PC aa C32:3	0.78	0.70	0.67	0.67	0.035	0.046	0.204	0.274
PC aa C34:1	182.8	203.8	225.9	268.1	12.236	<0.001	0.013	0.390
PC aa C36:0	2.6	3.0	2.9	3.6	0.197	0.036	0.013	0.487
PC aa C36:1	109.4	134.9	135.1	170.1	9.242	0.002	0.002	0.611
PC aa C36:3	104.3	111.8	114.4	129.0	5.596	0.018	0.053	0.525
PC aa C36:5	10.3	11.5	10.6	11.9	0.574	0.580	0.037	0.991
PC aa C38:3	77.4	91.7	83.1	98.3	5.038	0.227	0.005	0.939
PC aa C38:4	268.3	258.6	233.8	229.9	10.030	0.003	0.505	0.773
PC aa C38:5	59.3	58.6	52.2	54.0	2.260	0.012	0.808	0.568
PC aa C40:5	23.2	21.1	19.8	17.6	1.052	0.002	0.048	0.937
PC aa C42:4	0.75	0.65	0.69	0.61	0.032	0.125	0.007	0.803
PC aa C42:5	0.68	0.61	0.64	0.57	0.031	0.149	0.022	0.970
PC aa C42:6	0.75	0.67	0.71	0.65	0.033	0.388	0.045	0.730
PC ae C30:0	0.17	0.16	0.15	0.15	0.007	0.009	0.235	0.260
PC ae C30:1	0.30	0.20	0.17	0.14	0.038	0.013	0.095	0.433
PC ae C32:2	0.32	0.29	0.26	0.25	0.012	0.001	0.073	0.459
PC ae C36:4	17.2	16.0	14.4	13.5	1.307	0.050	0.411	0.901
PC ae C38:0	1.58	1.71	1.47	1.70	0.080	0.468	0.030	0.503
PC ae C40:1	1.25	1.40	1.20	1.45	0.067	0.968	0.004	0.432
PC ae C40:4	2.97	2.75	2.61	2.38	0.166	0.034	0.177	0.959
PC ae C42:3	0.32	0.39	0.34	0.44	0.024	0.131	0.001	0.578
PC ae C44:4	0.11	0.09	0.09	0.09	0.006	0.011	0.163	0.174

Data are presented as least-square means and pooled SEM. RFI, residual feed intake; FL, feed intake level. *n* = 7 per FL group, RFI rank, and sex; except for *n* = 8 high-RFI ad libitum females. RFI was calculated for the experimental period from 9 to 30 days post-hatch. ^1^ PC aa C, phosphatidylcholine with diacyl residue sum C; PC ae C, phosphatidylcholine with acyl-alkyl residue sum C.

**Table 4 metabolites-09-00038-t004:** Serum uric acid and cholesterol and proportion of lymphocytes in blood of low and high residual feed intake (RFI) broiler chickens fed either ad libitum or restrictively.

	Ad Libitum Feeding		Restrictive Feeding			*p*-Value		
Metabolite	Low RFI	High RFI	Low RFI	High RFI	SEM	FL	RFI	FL × RFI
Uric acid (mg/dL)	1.20	1.50	1.11	1.11	0.10	0.024	0.284	0.289
Cholesterol (mg/dL)	138	151	146	162	4.9	0.060	0.004	0.750
Lymphocytes (%)	84.5	83.1	87.1	88.4	1.85	0.038	0.966	0.476

Data are presented as least-square means and pooled SEM. RFI, residual feed intake; FL, feed intake level. *n* = 7 per FL group, RFI rank, and sex; except for *n* = 8 high-RFI ad libitum females. RFI was calculated for the experimental period from 9 to 30 days post-hatch.

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
