# Peer review of "Feed Restriction Reveals Distinct Serum Metabolome Profiles in Chickens Divergent in Feed Efficiency Traits"

_metabolites, 2019, doi:10.3390/metabo9020038_

Round 1

Reviewer 1 Report

The manuscript by Metzler-Zebeli et al. describes these authors’ efforts to identify metabolite signatures that reflect feed efficiency in broilers. This topic is relevant to the need to better understand the metabolic pathways that contribute to differences in feed utilization in broilers. Broilers with relatively high and low efficiency were selected and compared in the ad libitum and feed restricted states.  It is novel in comparing both low and high efficiency in the context of two different levels of intake, and it points to amino acids as a key set of metabolites that differ according to efficiency. The manuscript is fairly well-written and easy to read.  The data are thoroughly presented and discussed and support the authors’ conclusions.

The authors present a fairly thorough metabolomic profiling of chickens that differ in RFI, and in the fed and fasted state. The results are relatively descriptive, but the study is novel and is more of a discovery-based approach.  The results should be of interest to those in the poultry field, in particular.

Author Response

Dear reviewer,

Thank you very much for your comments.

Reviewer 2 Report

The manuscript showed that feed restriction reveals distinct serum metabolome profiles in chickens divergent in feed efficiency traits. The study is important for poultry science and discussion on metabolomics themselves seems good. Only one improvement is needed for the acceptance.

1)Authors used RFI (a metric for feed efficiency) as a key indicator. However, explanation of the RFI is not enough in the Introduction. For the readers who are not experts in animal nutrition, more explanation is necessary. For example, not only computation, but also advantages, disadvantages and limitations for the application in animal production.

Author Response

Dear Reviewer,

Thank you very much for your comments. We added Information regarding the RFI at the beginning of the Introduction section, lines 43-59 in the revised manuscript.